# Convolutional Neural Network Classification of Rest EEG Signals among People with Epilepsy, Psychogenic Non Epileptic Seizures and Control Subjects

**DOI:** 10.3390/ijerph192315733

**Published:** 2022-11-26

**Authors:** Michele Lo Giudice, Edoardo Ferlazzo, Nadia Mammone, Sara Gasparini, Vittoria Cianci, Angelo Pascarella, Anna Mammì, Danilo Mandic, Francesco Carlo Morabito, Umberto Aguglia

**Affiliations:** 1Department of Information Engineering, Infrastructure and Sustainable Energy (DIIES), University “Mediterranea” of Reggio Calabria, 89100 Reggio Calabria, Italy; 2Department of Science Medical and Surgery, University of Catanzaro, 88100 Catanzaro, Italy; 3Regional Epilepsy Center, Great Metropolitan Hospital “Bianchi-Melacrino-Morelli” of Reggio Calabria, 89100 Reggio Calabria, Italy; 4Department of Civil, Energy, Environmental and Material Engineering (DICEAM), University “Mediterranea” of Reggio Calabria, 89100 Reggio Calabria, Italy; 5Department of Electrical and Electronic Engineering, Imperial College London, London SW7 2AZ, UK

**Keywords:** EEG, empirical mode decomposition, convolutional neural network, deep learning, PNES, epilepsy

## Abstract

Identifying subjects with epileptic seizures or psychogenic non-epileptic seizures from healthy subjects via interictal EEG analysis can be a very challenging issue. Indeed, at visual inspection, EEG can be normal in both cases. This paper proposes an automatic diagnosis approach based on deep learning to differentiate three classes: subjects with epileptic seizures (ES), subjects with non-epileptic psychogenic seizures (PNES) and control subjects (CS), analyzed by non-invasive low-density interictal scalp EEG recordings. The EEGs of 42 patients with new-onset ES, 42 patients with PNES video recorded and 19 patients with CS all with normal interictal EEG on visual inspection, were analyzed in the study; none of them was taking psychotropic drugs before registration. The processing pipeline applies empirical mode decomposition (EMD) to 5s EEG segments of 19 channels in order to extract enhanced features learned automatically from the customized convolutional neural network (CNN). The resulting CNN has been shown to perform well during classification, with an accuracy of 85.7%; these results encourage the use of deep processing systems to assist clinicians in difficult clinical settings.

## 1. Introduction

Epilepsy is a neurological and medical disorder that affects millions of people each year, causing increasing socioeconomic health costs as well as decreased life quality and expectancy. It is a brain disorder characterized by recurrent, unprovoked seizures. The cause is frequently unknown, but it could be a brain injury or a genetic trait [1,2]. The starting point for diagnosing epilepsy is a careful examination of the symptoms, the collection of the history related to the health status of the affected person and the description of what happened in the moments immediately before the onset of the seizure, during its development and at its end. With this in mind, it is of paramount importance to acquire the testimonies of relatives, friends or bystanders who witnessed the episode. Due to the frequent absence of eyewitnesses and the high prevalence of seizure mimics (such as transient ischemic attacks, syncope, transient global amnesia, or vertigo), diagnosing epilepsy can be difficult. Moreover, when patients suffer from cognitive impairment, the diagnostic issue is much more challenging [3].

Epileptic seizures can also be confused with psychogenic non-epileptic seizures (PNES), i.e., events that semiologically mimic an epileptic seizure but lack the characteristic electrical discharge. PNES are characterized by the sudden and transient impairment of motor, sensory, autonomic, cognitive and/or emotional functions which are determined by psychological mechanisms and not by alterations in brain electrical activity. The psychological mechanisms underlying PNES are poorly understood and there is a lack of well-established, evidence-based treatments. They are common in neurological settings and are frequently associated with severe distress and disability [4]. Misdiagnosis may result in inappropriate and dangerous drug treatment.

PNES can be suspected clinically based on the clinical history and semiological features of the episodes, but it requires experienced clinicians and time-consuming interviews. It needs to be confirmed by a video-EEG recording of typical episodes, which is the gold standard for identifying this type of seizure [5,6].

The main diagnostic test for both PNES and ES is the EEG. It is a non-invasive and low-cost method of recording the electrical activity of the brain with high temporal resolution [7]. Recording an ES or PNES during the screening is uncommon. During the examination, the subject under analysis may be asked to do something that might trigger seizures such as, for example, breathing fast or staring at flashing light sources, but this method poses ethical problems [6]. That is why interictal EEG on visual inspection may appear normal, but advanced analyses using engineered methods are able to pick up hidden alterations so as to allow large-scale diagnosis without the ethical problems mentioned above.

In this regard, several automatic classification algorithms based on EEG have been proposed in the literature [8,9,10,11,12] and a significant amount of research has focused on the diagnosis of ES and PNES, but it is still an open and a challenging issue. The remarkable technological advances of deep learning (DL) algorithms have provided a boost in solving the task. They have abilities to learn complex patterns, including latent features that are not discernible by visual inspection or standard approaches. Indeed, in a previous paper, we [13] compared the processing capabilities of standard algorithms with deep algorithms. We analyzed 18 patients with ES and 18 patients with PNES. In this study, we extend the dataset and include CS in the classification. This increases the flexibility of the network in distinguishing the features of potentially healthy subjects as well as giving more clinically useful information. In addition, unlike [13], we use EMD instead of wavelets. It is a fully data-driven, adaptive technique that extracts the oscillatory modes present in the data, yielding a variable number of components. EMD overcomes the limitations of the wavelet approach. In fact, in EMD, the basis for the analysis is generated by the same analyzed signal; in other words, it is not necessary to set the level of decomposition a priori.

Thus, in this paper, EEGs of 103 subjects (42 ES, 42 PNES, 19 CS) were collected at the Regional Epilepsy Centre, Great Metropolitan Hospital of Reggio Calabria, University of Catanzaro and an automatic elaboration pipeline was developed based on the classification of the EMD-decomposed signal using DL. Specifically, in this study, we propose a data-driven convolutional neural network (CNN) capable of differentiating intrinsic mode functions (IMFs), i.e., components of empirical mode decomposition (EMD), extracted from the EEGs of ES, PNES and CS. The EMD performs time-frequency analysis while remaining in the time domain.

Based on these assumptions, the following contributions are achieved by this work:Data-driven DL approach based on convolutional neural network (CNN) to classify ES, PNES and CS subjects by analyzing only noninvasive EEG recordings;Automated system potentially useful for clinical applications.

## 2. Materials and Methods

We propose an analytical model of EEG segments based on machine learning that can be used to classify three different classes of subjects. To process them, the decomposition of the EEG signal into appropriate function bases, i.e., empirical mode decomposition (EMD) combined with machine learning methods, is proposed.

The workflow of the method is graphically represented in Figure 1: (1) acquisition of the 19-channels EEG recording; (2) segmentation of the EEG signals into 50% overlapped EEG segments; (3) estimation of IMFs in the range 1–3 for each EEG channel. The IMFs are used as input to a customized convolutional neural network characterized by two convolution layers, two max pooling layers, and three fully connected layers followed by a softmax layer that performs three-way (ES vs. PNES vs. CS) classification; (4) Testing the classification of the IMFs (i.e., the corresponding EEG segments) as belonging to ES, PNES or CS subjects, is tested (EMD segment-based classification).

### 2.1. Study Population

EEGs of 103 subjects previously enrolled at the Regional Epilepsy Centre, Great Metropolitan Hospital of Reggio Calabria, University of Catanzaro, Italy were used: 42 subjects with new-onset clinically diagnosed ES, 42 subjects with video-recorded PNES, and 19 CS clinically healthy on EEG inspection. Only artifact-free EEG recordings were considered.

### 2.2. EEG Recording

The EEG signals were acquired using a standard 10–20 system [14] with 19 channels. The inclusion criteria were willingness to participate, informed consent, and normal interictal EEG at visual inspection. The chronic use of psychotropic drugs at the time of registration was an exclusion criterion. The participant was informed and instructed on the diagnostic setup while seated in a hushed, dimly lit room during the recording process.

### 2.3. EEG Processing

Following the acquisition, the EEGs were visually inspected by an expert clinician for artifact removal, subsampled at 256 Hz and archived in MAT (MATLAB^®^ File Format) file. The EEG recording was then band-pass filtered at 0.5–32 Hz using the Matlab toolbox EEGLab [15,16] and partitioned into 5-second EEG segments (1280 samples derived from 5(s) × 256(Hz)) overlapping by 50% in order to improve dataset extension with data augmentation. Three well-behaved components defined as Intrinsic Mode Functions (IMFs) were then retrieved by using algorithms based on empirical mode decomposition (EMD).

### 2.4. Empirical Mode Decomposition

Empirical mode decomposition (EMD) is a fully data-driven adaptive technique for obtaining the oscillatory modes present in data, resulting in a variable number of components. Huang et al. first proposed it [17].

EMD begins by estimating a signal locally as a sum of a local trend and a detail signal component: the local trend is a low-frequency component, and the local detail reports high frequencies. The high-frequency (detail) components of EMD are referred to as intrinsic mode function (IMF), while the low-frequency component is referred to as residual. The procedure is then applied to the residual again, this time as a new time series, yielding a new IMF and residual. As a result, the IMFs can be extracted iteratively. The IMFs should meet two conditions:The number of extrema and zero crossings in the entire data set must be equal or differ from each other by no more than one;At every point, the mean value of the envelope defined by the local maxima and the envelope described by the local minima must be zero.

The conditions listed above are empirical, as there is no explicit formula for calculating IMFs. As it will be demonstrated, the analysis of the power spectrum of IMFs allows us to confirm that these functions represent the original signal decomposed into different time scales or frequency bandwidths [17,18]. Given a signal *x*(*t*), the EMD’s effective algorithm is as follows [17]:1.Identify all extrema (maxima and minima) of *x*(*t*);2.Generate the upper and lower envelope ($e_{min}$(*t*), $e_{max}$(*t*)) by connecting the maxima and minima points separately with cubic spline;3.Compute the local mean *r*(*t*) = ($e_{min}$(*t*) + $e_{max}$(*t*))/2;4.Extract the detail *d*(*t*) = *x*(*t*) − *r*(*t*);5.Iterate on the residual *r*(*t*).

At the end of the decomposition process, the EMD method expresses the signal *x*(*t*) as the sum of a finite number of IMFs and a final residual [17]:(1)$$\;x\left(t\right)=\sum_{i=1}^{n}h_{i}\left(t\right)+r_{n}\left(t\right)$$
where $h_{i}\left(t\right)$ are the IMFs and $r_{n}\left(t\right)$ is a final residual, which is less than an arbitrarily predetermined threshold. The algorithm iteratively works by identifying the signal’s extrema and breaking it down, ensuring that the number of modes is finite. At each iteration, the envelope is estimated by interpolating the signal’s extrema at each iteration. To avoid over-sampling issues, the EMD algorithm must focus on both the choice of extrema and the boundary conditions for the analysis of discrete-time sequences. IMFs provide the original signal with a complete and “nearly” orthogonal basis. Different components may have parts with similar frequencies at different time durations in some conditions, but locally, any two components will tend to be orthogonal. The EMD approach has gained popularity in a variety of fields of applications [19,20,21,22,23]. It is frequently used in contrast to the wavelet approach [24].

### 2.5. Proposed Architecture

The proposed CNN architecture has been chosen empirically after several experimental tests. It is intended to accept fixed-size EEG segments of 19 × 1280 × 3 (where 19 denotes the number of EEG channels, 1280 denotes the number of samples, and 3 denotes the number of IMF). After several experimental tests, also the number of filters and their size were determined empirically. We used 64 learnable filters of size 1 × 6. Each filter convolves with the input image with a stride s = 1 × 2, developing 64 feature maps with a 50% reduction in input size 19 × 640 × 64. The CNN now has 1216 learnable parameters. The max pooling layer (MaxPool) reduces the size of the features maps from the convolutional layer’s initial size of 19 × 640 × 64 to 19 × 320 × 64 using 1 × 2 filters with a stride = 1 × 2. The next convolutional layer is made up of 32 learnable filters with a size of 1 × 3. Each filter convolves with the features maps with a stride s = 1 × 2, producing 32 feature maps with a half-size input: 19 × 160 × 32. The CNN now contains 6.176 parameters that can be learned. The second max pooling layer (MaxPool), which reduces the size of the features maps, comes after the convolutional layer and reduces the features maps size from 19 × 160 × 32 to 19 × 80 × 32 using 1 × 3 filters with a stride = 1 × 2. The collected feature maps were flattened using the Flatten layer, resulting in 48.640 features, which were then fed as input to two fully connected layers, the first with 128 hidden units and the second with 32 hidden units. It is followed by a Dropout layer (of 0.3), which separates from the third fully connected layer with 16 hidden neurons in order to improve generalization and prevent overfitting. A soft-max (SF) layer that estimates the three-way classification is added at the bottom of the network. The total number of parameters utilized is shown in Table 1. Different training options were evaluated for CNN: varying the learning rate α, the rate of decay of the first moment $\beta_{1}$ and the rate of decay of the second moment $\beta_{2}$. It was observed that the best results were obtained with: learning rate $\alpha =10^{-2}$, first-moment decay rate $\beta_{1}$ = 0.9 and second-moment decay rate $\beta_{2}$ = 0.999 according to the practical recommendations reported in [25,26].

We performed a k-fold with k = 5 to avoid overfitting generated by data obtained from different subjects with several EEG segments per subject.

### 2.6. Metrics of Classification

The study’s objective was to evaluate the network’s ability to correctly classify the class of EEG segments (ES vs PNES VS CS). The accuracy of EEG segments classification was assessed using the following metrics:(2)$$\;ACCURACY=\frac{TP+TN}{TP+TN+FP+FN}\;$$
(3)$$\;PRECISION=\frac{TP}{TP+FP}\;$$
(4)$$\;RECALL=\frac{TP}{TP+FN}\;$$
(5)$$\;F1-score=2\times \frac{PRECISION\times RECALL}{PRECISION+RECALL}\;$$
where *TP*, *TN*, *FP* and *FN* represent the true positive, true negative, false positive and false negative, respectively. Specifically, *TP* and *TN* are the numbers of EEG segments classified correctly; *FP* is the number of EEG segments incorrectly classified, *FN* is the number of EEG segments misclassified.

## 3. Experimental Results

The goal was to assess the ability of the network to correctly classify IMFs of EMD extracted from the EEG segments as belonging to class ES, PNES or CS (epoch-based classification).

The dataset included 103 EEG recordings. EEG segments of 5 s were extracted and preprocessed to construct the dataset of IMFs that was used to train and test the network.

The ability to correctly classify ES, PNES or CS EEG segments was evaluated through standard metrics: accuracy, precision, recall and F1-score as outlined in Table 2.

Given that the entire dataset contains 25.754 EEG segments, 80% of epochs were used for training and the remaining 20% of epochs were used for testing. Each division of the dataset into train and test considered balanced portions with the class of subjects, in the entire 5-fold iteration. The ability to correctly classify ES, PNES or CS EEG segments was evaluated through standard metrics: accuracy, precision, recall and F1-score. This is an excellent result, given that interictal EEGs are difficult to analyze, as confirmed by the few papers in the literature. Table 3 summarizes the results of classification algorithms of ES and PNES from interictal EEG present in the literature. From the comparison, the result confirmed the effectiveness of the proposed approach compared with conventional classifiers based on handcrafted feature extraction of EEG signals. EMD, in fact, seems to contain more information than manual features extracted directly from EEG, even if relevant. EMD allows a signal to be decomposed into physically significant components with multi-resolution, thus it succeeds in containing more information. Specifically, the best classification performances were observed in the binary classification performed in previous work. However, considering the 3-way classification, we can be completely satisfied with the result, given the more difficult classification and more importantly, given the clinical utility as our primary goal.

## 4. Discussion

In this paper, we propose EEG data DL classification methods for classifying EEG recordings from ES, PNES, and CS by interictal EEG, using a processing pipeline that includes EMD and advanced artificial intelligence methodologies. Our proposed model has been shown to provide good classification accuracy for EEG data. We use EMD for signal decomposition. EMD is completely data-driven and performs multi-scale decomposition and temporal time analysis of real-world signals into physically meaningful components. The development of an innovative method based on the decomposition of EEG segments using EMD combined with DL techniques was one of the study’s main contributions. Compared to the previous work [13] using wavelets, we have several advantages because EMD decomposes the original signal without a preselected basis function, as it is instead necessary for wavelet decomposition. EMD overcomes this limitation by generating the algorithm’s basis from the same analyzed signal using a data-driven heuristic procedure [24,28]. In EMD, the frequency is calculated by differentiation instead of convolution, as in wavelet analysis; this allows the volatility principal limitations to be overcome. The disadvantage of EMD is that it lacks a theoretical foundation; instead, it is purely based on empirical evidence. However, since the wavelet can be influenced by mother functions, the EMD algorithm overcomes it. EMD has also been preferred over other modal decomposition algorithms, e.g., over the VMD method, a non-recursive decomposition technique in which modes are decomposed simultaneously with respect to their center frequency [29], because EMD has the advantage of decomposing the signal adaptively and, in a data-driven way.

DL, on the other hand, is an advanced artificial intelligence technique capable of automatically learning latent discriminating variables (features) from input data, avoiding the need for hand-crafted feature extraction. In particular, CNN is a DL classifier that is widely regarded as being at the cutting edge of this field. We also investigated the effect of network depth on classification time. Our experimental results indicate that proper depth balance of architectures are crucial for accurate classification. Based on our findings, we recommend using three *Fully Connected* hidden layers with 128-32-16 hidden units per layer for this EEG data classification tasks.

To our knowledge, this is the first study to address the classification of ES, PNES, and CS using a data-driven framework based on EMD and DL.

In the previously introduced CNN classifier, we differentiate EEG recordings of only ES and PNES subjects [13]. The system was built across wavelet signal decomposition and automatic feature extraction from time representations and EEG time series. Although good classification performance was obtained (94.4% accuracy in binary ES vs. PNES classification), it was tested on a small data set (36 subjects) across two classes. Three classes amounting to 103 subjects were examined in this study. The proposed DL procedure necessitated a greater number of EEGs to perform a more difficult classification based on three classes rather than two.

The 50% overlap for data augmentation, combined with the large extension of the data, allowed us to significantly expand the dataset, resulting in a total of 35 h, 46 min, and 10 s database, large enough to train a CNN with convolutional, Relu, and Max Pooling Layer (CNN) modules with excellent generalization capabilities.

The proposed CNN performed exceptionally well despite the inclusion of a new class that significantly complicates the processing outcome.

In CNN, there were more than 6 million learning parameters, and the computation required 3946 s. It should be noted that the study’s findings led to a classification latency time of 31 s, which is still a perfectly reasonable waiting time but prevents the definition of a real-time system. This is due to the increase in computational complexity (and, consequently, computational cost) in the study.

DL methods might be a significant advance in practical clinical applications. The suggested approach, however, has several drawbacks. The number of subjects is the key constraint; future work could have precisely balanced classes to minimize bias. These results may influence clinical practice. Indeed, the proposed classification system may help in the differential diagnosis between the three classes and particularly between ES and PNES. The epoch-based approach, however, does not allow to perform of clear-cut discrimination on a patient basis, but it may help the clinician by providing a probability of classification for the single EEG trace. However, despite improved classification algorithms, misclassifications may still occur because of tiny changes between patients that result from interictal recordings.

## 5. Conclusions

In this paper, the authors introduced a new EMD-based DL method for classifying EEG segments of subjects with ES, PNES or CS. The originality of the proposed method lies in using EMD for the automatic extraction of multiresolution features to be sent as input to a convolutional neural network (CNN) capable of automatically mapping latent features from interictal EEG segments.

The use of interictal EEG is another important contribution to the originality of the paper, which increases the complexity of the study but has completely undisputed clinical advantages.

Experimental results indicate that CNN provided high performance in classifying EEG segments epochs, which is not achieved by standard learning algorithms given the lack of handcrafted feature extraction. Future training and testing processes for the proposed EMD-based CNN architecture will be optimized utilizing more potent graphics processing units (GPUs), enabling real-time analysis and a patient-based approach. To further illustrate and make use of the generalization potential of DL approaches in clinical applications, a larger cohort of ES/PNES/CS participants will be taken into consideration. On the strength of an information theoretic approach, classifications on various IMFs will also be analyzed to understand which part of the data has more information.

## Figures and Tables

**Figure 1 ijerph-19-15733-f001:**
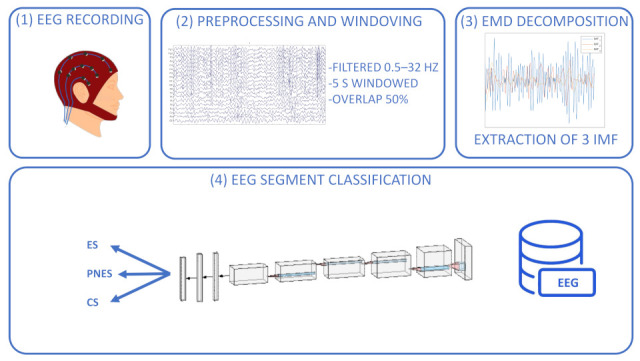
The electroencephalogram (EEG) was recorded and saved on a computer. Following that, the EEG was cleaned of artifacts, filtered, and divided into 50% overlapped EEG segments of 5 s each. Three IMFs of EMD over each channel were estimated for each EEG segment. The obtained database was the input of a convolutional neural network composed of two convolutional layers (+ReLu activation layer), two max pooling layers, three fully connected layers separated by a dropout layer, and a sigmoid layer that performs classification tasks (three ways: ES, PNES vs. CS).

**Table 1 ijerph-19-15733-t001:** The proposed CNN architecture’s total number of parameters. The CNN consists of two convolutional layers (+ReLu), two max pooling layers, three fully connected layers, one dropout layer, and a sigmoid layer that performs three-way classification tasks.

Layer Name	Output Shape	Parameters
Input	19 × 1280 × 64	
1st Conv2D	19 × 640 × 64	1216
2nd MaxPooling	19 × 80 × 32	
Flatten	48.640	
1st Dense	128	6.226.048
2nd Dense	32	4128
Dropout	32	
3rd Dense	16	528
Output	3	51
Total		6.328.147

**Table 2 ijerph-19-15733-t002:** The proposed CNN’s classification metrics in terms of mean value.

Accuracy	Precision	Recall	F1 Score
85.7%	85.7%	85.7%	85.7%

**Table 3 ijerph-19-15733-t003:** Comparison of accuracy results with ES vs. PNES classifications proposed in the literature.

References	Classification Type	Classifiers	Accuracy
[27]	Binary classification of ES vs. PNES based on coverage feature of the EEG microstate analysis in beta-band	Random Forest SVM (Linear) SVM (RBF) Decision Tree kNN Gradient Boost	79.4% 66.8% 65.6% 76.4% 77.6% 78.8%
[13]	Binary classification of ES vs. PNES in interictal EEG	CNN MLP SVM (Linear) LDA QDA	94.4% 47.2% 58.3% 55.6% 55.6%

## Data Availability

Not applicable.

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
