# Peer review of "Convolutional Neural Network Classification of Rest EEG Signals among People with Epilepsy, Psychogenic Non Epileptic Seizures and Control Subjects"

_ijerph, 2022, doi:10.3390/ijerph192315733_

Round 1
Reviewer 1 Report
The major concerns about the article are:
1) According to subsection 2.1 of the article, it is believed that the authors collected data from subjects directly. However, the article does not mention the approval of the study by the ethics committee to carry out experiments on humans.
2) Section 3 of the article briefly presents the experimental results. It is not clear how these results are sufficient to support the objectives to be achieved. Perhaps a Table 3 comparing the results of the on-screen study with the main state-of-the-art works is needed.
3) Although the authors have presented an interesting discussion of the results, the article does not present the conclusions section highlighting the main contributions of the research.
Reviewer 2 Report
In this paper, a DL classification method for EEG data is proposed, which is used to classify EEG records from ES, PNES and CS by intermittent EEG, using a processing pipeline including EMD and advanced artificial intelligence methods. The proposed model has proved that EEG data provides good classification accuracy.
This is a good work and has high clinical experimental significance, but there are some problems.
1. In the part of materials and methods, the author puts forward a combination of EMD and machine learning method to analyze EEG segments, which is a good point, but why not try machine learning classification model for EEG data classification.
2. EMD modal decomposition is a signal processing algorithm in the last century, not a machine learning algorithm. Why not try more other new modal decomposition algorithms, such as VMD and DMD, and what advantages does it have?
3. Why is there no comparative experiment with the most advanced convolution neural network?
4. Whether the author has opened up data sets and code to facilitate more researchers to reproduce the research, what is the parameter design for the CNN model.
Round 2
Reviewer 1 Report
After the review carried out by the authors, I believe that the article is in conditions to be approved in its present form.